

# Introgressive hybridization in a Spiny-Tailed Iguana, *Ctenosaura pectinata*, and its implications for taxonomy and conservation

Eugenia Zarza[1,2,3], Víctor H. Reynoso[1], Christiana M. A. Faria[4,5] and Brent C. Emerson[6]

[1] Departamento de Zoología, Instituto de Biología, Universidad Nacional Autónoma de México, Ciudad de México, Mexico
[2] Grupo Académico de Biotecnología Ambiental, El Colegio de la Frontera Sur, Unidad Tapachula, Tapachula, Chiapas, Mexico
[3] CONACYT, Ciudad de México, Mexico
[4] School of Biological Sciences, University of East Anglia, Norwich, UK
[5] Current Affiliation: Departamento de Biologia, Universidade Federal do Ceará, Campus do Pici, Fortaleza, Ceará, Brasil
[6] Island Ecology and Evolution Research Group, Instituto de Productos Naturales y Agrobiología (IPNA-CSIC), C/Astrofísico Francisco Sánchez 3, La Laguna, Tenerife, Canary Islands, Spain

Corresponding authors
Eugenia Zarza,
eugenia.zarza@gmail.com
Víctor H. Reynoso,
vreynoso@ib.unam.mx

## ABSTRACT

Introgression, the transmission of genetic material of one taxon into another through hybridization, can have various evolutionary outcomes. Previous studies have detected signs of introgression between western populations of the Mexican endemic and threatened spiny-tailed iguana, *Ctenosaura pectinata*. However, the extent of this phenomenon along the geographic distribution of the species is unknown. Here, we use multilocus data together with detailed geographic sampling to (1) define genotypic clusters within *C. pectinata*; (2) evaluate geographic concordance between maternally and biparentally inherited markers; (3) examine levels of introgression between genotypic clusters, and (4) suggest taxonomic modifications in light of this information. Applying clustering methods to genotypes of 341 individuals from 49 localities of *C. pectinata* and the closely related *C. acanthura*, we inferred the existence of five genotypic clusters. Contact zones between genotypic clusters with signatures of interbreeding were detected, showing different levels of geographic discordance with mtDNA lineages. In northern localities, mtDNA and microsatellites exhibit concordant distributions, supporting the resurrection of *C. brachylopha*. Similar concordance is observed along the distribution of *C. acanthura*, confirming its unique taxonomic identity. Genetic and geographic concordance is also observed for populations within southwestern Mexico, where the recognition of a new species awaits in depth taxonomic revision. In contrast, in western localities a striking pattern of discordance was detected where up to six mtDNA lineages co-occur with only two genotypic clusters. Given that the type specimen originated from this area, we suggest that individuals from western Mexico keep the name *C. pectinata*. Our results have profound implications for conservation, management, and forensics of Mexican iguanas.

## INTRODUCTION

The role of introgression, or gene flow between divergent lineages (*Streicher et al., 2014*; *Haenel, 2017*; *Kumar et al., 2017*; *Pilot et al., 2018*) in shaping biodiversity is receiving increasing attention in different taxa and geographic areas (*Abbott et al., 2013*; *Haus, Roos & Zinner, 2013*; *Mallet, Besansky & Hahn, 2016*). There is evidence suggesting that introgression can increase the risk of extinction in endangered species through genetic swamping (*Frankham, 2006*). Additionally, introgression can have deleterious effects in hybrids, lead to adaptation by the emergence of novel genotypes, or have no effect on the fate of a species (*Seehausen, 2004*; *Mallet, 2005*; *Frankham, 2006*; *Kronforst, 2012*; *Pardo-Diaz et al., 2012*). Given these various outcomes, it is particularly important to study the extent and impact of introgression in biologically rich areas like Mesoamerica, where general patterns of genetic diversity are just starting to be uncovered (*Ornelas et al., 2013*; *Mastretta-Yanes et al., 2015*; *Nieto-Montes de Oca et al., 2017*; *Bryson et al., 2017*; *Rodríguez-Gómez & Ornelas, 2018*). The results of such studies can have direct implications for species delimitation and, ultimately, conservation and wildlife management (*Gompert, 2012*).

The dry tropical forests of the western lowlands of Mexico are part of the Mesoamerica Hot Spot (*Myers et al., 2000*). Many phylogeographic studies have focused on this area, though only a few of them have employed a multilocus approach that can detect the presence of introgression (*Daza et al., 2009*; *Greenbaum, Smith & De Sá, 2011*; *Pringle et al., 2012*; *Arbeláez-Cortés, Milá & Navarro-Sigüenza, 2014*; *Arbeláez-Cortés, Roldán-Piña & Navarro-Sigüenza, 2014*). In the spiny-tailed iguana *Ctenosaura pectinata*, distributed in the lowlands of the Pacific slope and the Balsas Depression in Mexico (*Smith & Taylor, 1950*; *Köhler, Schroth & Streit, 2000*), initial phylogeographic studies recovered eight mitochondrial DNA (mtDNA) lineages, recognized as statistically supported nodes in a phylogeny: North A, North B, North C, Colima, Balsas, Guerrero, Oaxaca and South (Fig. 1; Fig. S1; *Zarza, Reynoso & Emerson, 2008*). *Ctenosaura acanthura,* found in the lowlands of the Gulf of Mexico, appeared as sister to the South lineage, whereas *C. hemilopha* and *C. similis* were recovered as clearly distinct lineages (*Zarza, Reynoso & Emerson, 2008*).

Genetic distances (*Tamura & Nei, 1993*) between *C. pectinata* mtDNA lineages range from 4.11% to 11.57%, similar to values estimated among species of iguanas of the genus *Cyclura* (*Malone et al., 2000*). The North and Colima mtDNA lineages show the largest distance measured within *C. pectinata* (*Zarza, Reynoso & Emerson, 2008*). This phylogeographic break occurs in the vicinity of the Trans-Mexican Volcanic Belt (TMVB; Fig. 1), on the central western coast of Mexico and, probably occurred between 1.1 and 3.1 million years ago (*Zarza, Reynoso & Emerson, 2008*). This geological feature, a volcanic belt that covers central-southern Mexico from the Pacific Ocean to the Gulf of Mexico, has attracted many biogeographers because this represents the distributional

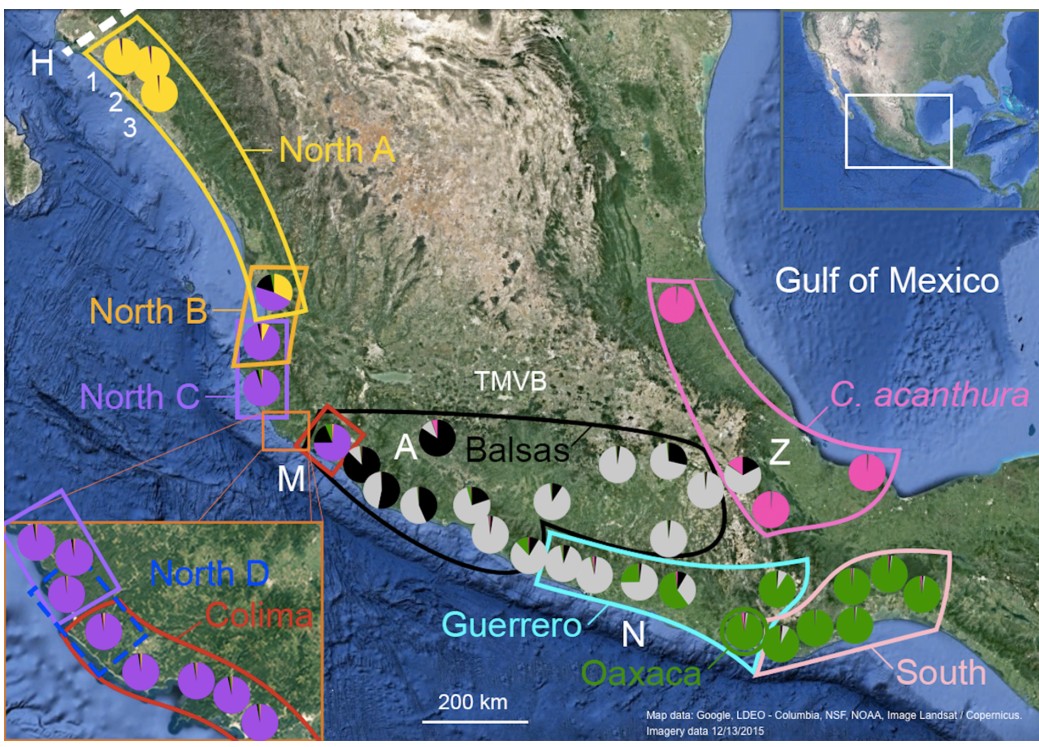

**Figure 1 Geographic distribution of mtDNA lineages and genotypic clusters within *Ctenosaura pectinata* and *C. acanthura*.** Lines represent the geographical limits of the mtDNA lineages. Colors and lineage names follow the scheme shown in the haplotype network (Fig. S1). Pie charts show proportion of ancestry among individuals sampled in each locality with colors equivalent to STRUCTURE clusters. A, Apatzingán; H, contact zone between *C. hemilopha* and *C. pectinata*; M, Manzanillo; N, Las Negras; Z, Zapotitlán de las Salinas. Numbers 1–3 show site locations mentioned in the main text where new mtDNA haplotypes were uncovered. Map was generated with Google Earth (Data LDEO—Columbia, NSF, NOAA, Image Landsat/Copernicus. Imagery date 12/13/2015).

limits of highland and lowland taxa (*Mastretta-Yanes et al., 2015*; *Zaldivar-Riverón, Leon-Regagnon & Nieto-Montes de Oca, 2004*; *Devitt, 2006*; *Mulcahy, Morrill & Mendelson, 2006*; *Bryson, García-Vázquez & Riddle, 2012*; *Blair et al., 2015*). Additional multilocus data and detailed geographic sampling of *C. pectinata* in this area revealed a ninth mtDNA lineage occurring between North C and Colima lineages: North D (*Zarza, Reynoso & Emerson, 2011*; Fig. 1). Interestingly, the North C, North D, Colima and Balsas mtDNA lineages show geographically discordant patterns with two clusters identified with microsatellite markers (*Zarza, Reynoso & Emerson, 2011*). The discordance likely resulted from contemporary and/or past introgression among lineages, coupled with male sex biased dispersal (*Zarza, Reynoso & Emerson, 2011*). It is unknown if geographic discordance between mtDNA and microsatellite markers, and introgression are restricted to this part of *C. pectinata* distribution, or if it is prevalent among other neighboring populations.

These previous molecular studies in *C. pectinata* have uncovered diversity that had been overlooked or not detected by the most recent morphological revisions of the species and closely related taxa (*Köhler, Schroth & Streit, 2000*; *Köhler, 2002*). This is in contrast to

earlier studies of the genus. *Ctenosaura pectinata* was described by *Wiegmann (1834)*. *Bailey (1928)*, in a revision of the genus recognized five species (*C. brachylopha, C. pectinata, C. acanthura, C. brevirostris, C. parkeri*) within the range of what we currently know as *C. pectinata*. He stated that *C. acanthura* was the most widely distributed, inhabiting both the western and eastern coasts of Mexico. He indicated that *C. pectinata* and *C. brevirostris* had approximately the same distribution on the western foothills of Mexico, with "Colima" as type locality. *Ctenosaura parkeri* was only known from two collecting sites in Jalisco and Nayarit. *Ctenosaura brachylopha* was described as inhabiting the northern states of Nayarit and Sinaloa. Without giving any justification, *Smith & Taylor (1950)* lumped *C. brachylopha, C. brevirostris* and *C. parkeri* with *C. pectinata* and restricted the name *C. acanthura* for iguanas from the Gulf of Mexico area.

In light of recent molecular studies and previous morphological classifications, revisiting *C. pectinata* genetic diversity and taxonomy is warranted. Taxonomic modifications should rely on morphological information, a multilocus approach, and comprehensive geographic sampling (*Leaché & Fujita, 2010*; *Bauer et al., 2011*; *Rittmeyer & Austin, 2012*). A multilocus approach facilitates the identification of genotypic clusters: groups of individuals that have few or no intermediates when in contact (*Mallet, 1995*). Such groups may inter-grade freely at their boundaries, but be strongly differentiated and relatively conserved in morphology, genetics and ecology. This implies that species can be affected by gene flow, selection and history, but they are not necessarily defined by these processes (*Mallet, 1995*). Defining genotypic clusters is useful in cases where gene flow between otherwise differentiated clusters occurs, for example in contact zones, as might be the case of *C. pectinata*.

Here, we use multilocus data from individuals sampled across the ranges of *C. pectinata* and the closely related *C. acanthura*. Our specific aims are to: (1) define genotypic clusters; (2) investigate the levels of geographic concordance between mtDNA lineages and genotypic clusters; (3) evaluate evidence for introgression between clusters; and (4) re-define taxonomic entities based on maternally and biparentally inherited markers, and compare these to previous taxonomic hypothesis.

## MATERIALS AND METHODS
### Sampling and laboratory procedures
Spiny-tailed iguanas were collected between 2004 and 2006 using tomahawk traps, noosing, or by hand within the recognized distribution of *C. pectinata* and *C. acanthura*. The narrow area of sympatry between *C. pectinata* and *C. hemilopha* in northern Mexico was excluded to avoid the inclusion of *C. hemilopha* alleles in the analyses (*Zarza Franco, 2008*; Fig. 1). All samples have been analyzed in previous studies (*Faria, 2008*; *Zarza, Reynoso & Emerson, 2008*, *2011*, *2016*; *Zarza Franco, 2008*; *Faria et al., 2010*;) to obtain microsatellite and/or mtDNA data (see File S1 for details). However, all these genetic data have not been analyzed together. All data is available from GenBank (File S1) or as supplementary material in *Zarza et al. (2016)*, including two previously unpublished mtDNA sequences (GenBank accession numbers KT003209–KT003210), and microsatellite data (File S1) from three localities in northern Mexico (Fig. 1, sites 1–3).
We generated datasets for both type of markers that are mostly overlapping regarding sample content (microsatellite $n = 341$, mtDNA $n = 344$) with 317 individuals out of 368, represented in both datasets. This study comprises samples from 53 out of 67 localities sampled in the above-mentioned studies; individuals from 49 localities were included in the microsatellite dataset. In some instances, individuals failed to amplify for mtDNA in earlier studies, but were successfully genotyped (24 out 341 samples; File S1). All mtDNA lineages described in previous publications were represented in the mtDNA dataset analyzed herein.

A thorough description of the sampling and laboratory methods can be found in *Zarza, Reynoso & Emerson (2008*, *2011*, *2016)*, *Faria et al. (2010)*; however a summarized version follows. From each individual, a tail clip, or a 0.15 µl blood sample from the caudal vein were taken and preserved in ethanol. DNA samples were purified using a modified salt precipitation protocol (*Aljanabi & Martinez, 1997*). A total of 561 bp fragment of the mitochondrial NADH dehydrogenase, subunit 4 (ND4) gene was amplified via polymerase chain reaction (PCR) and sequenced using primers ND4, ND4Rev (*Arèvalo, Davis & Sites, 1994*), ND4F1 (*Zarza, Reynoso & Emerson, 2008*) and ND4R623 (*Hasbún et al., 2005*) with conditions described by *Zarza, Reynoso & Emerson (2008)*. Individuals were genotyped with nine microsatellite markers. Loci Cthe12, Cthe37 (*Blázquez, Rodríguez Estrella & Munguía Vega, 2006*), Pec01, Pec03, Pec16, Pec20, Pec25, Pec73 and Pec89 (*Zarza et al., 2009*), were PCR amplified using conditions described by *Zarza, Reynoso & Emerson (2011)* and run in two multiplexes that allow for loci separation by color and size in an automated ABI PRISM® 3730 (Applied Biosystems, Foster City, CA, USA). Fragment size was visualized with the GeneMapper software version 4.0 (Applied Biosystems, Foster City, CA, USA).

The School of Biological Sciences Ethical Review Committee at the University of East Anglia approved this study as stated in an "Approval letter" to EZ. All efforts were made to minimize stress when taking blood samples, which were obtained under the permits SEMARNAT SGPA/DGVS/08239, SGPA/DGVS/02934/06, 03563/06 to VHR.

## Data analyses

### Mitochondrial DNA data

A median joining haplotype network was calculated with Network (*Bandelt, Forster & Rohl, 1999*) to update previously proposed haplotype networks (*Zarza, Reynoso & Emerson, 2008*, *2011*). SAMOVA 2.0 (*Dupanloup, Schneider & Excoffier, 2002*) was used to define groups of populations that are geographically homogeneous and maximally differentiated from each other and to estimate their hierarchical differentiation. A total of 100 initial independent processes were tested followed by 10,000 steps of the simulated annealing process, which maximizes the proportion of total genetic variance among groups. Previous studies, uncovered nine mtDNA clades (*Zarza, Reynoso & Emerson, 2008*, *2011*). To test this grouping pattern and to explore if a larger number of groups existed, SAMOVA analyses were run under scenarios of two to 15 groups ($K$) without geographic restrictions. No coherent geographic structure was detected at $K = 15$, thus higher values of $K$ were not tested. The $F_{CT}$ index (proportion of total genetic

variance due to differences between groups of populations) was used to select the best grouping, that is, the most suitable $K$. This index reflects the among-group component of the overall genetic variance. We selected the number of groupings that maximizes this component, meaning that under that scenario the groups of populations are maximally differentiated from each other (*Dupanloup, Schneider & Excoffier, 2002*). To accomplish this, the most suitable $K$ value was selected based on the observed changes of $F_{CT}$ among consecutive $K$ values. We considered arbitrarily that the most suitable value of $K$ is observed when there is a $F_{CT}$ change <1% between two consecutive $Ks$. We refer to this as $\Delta F_{CT}$ obtained as $F_{CT\ K+1} - F_{CT\ K}$, reflecting changes in the percentage of variation explained by $F_{CT}$. Bar plots were created with R 2.15 (*R Core Team, 2012*) to show the mtDNA lineage of each individual as determined by the haplotype network (Fig. S1), and to illustrate the results of SAMOVA.

### Microsatellite data

The software GENEPOP 4.1 (*Rousset, 2008*) was used to estimate allele and null allele frequencies, to perform tests for linkage disequilibrium between pairs of loci and to detect deviations from Hardy–Weinberg equilibrium. $F_{ST}$ values between localities were calculated with Arlequin 3.5 with the pairwise differences distance method (*Excoffier & Lischer, 2010*).

The possible number of genotypic clusters under a scenario of admixture was inferred with STRUCTURE 2.3.2 (*Pritchard, Stephens & Donnelly, 2000*). Simulations were run assigning a uniform prior for the parameter Alpha (degree of admixture) and estimating the allele frequency parameter (Lambda) assuming correlated allele frequencies. Previous studies in *C. pectinata* showed that a $K > 10$ was unlikely (*Zarza, Reynoso & Emerson, 2016*), thus we limited our STRUCTURE analyses to $K = 2$–$K = 10$, with 10 iterations for each value and 10 million Markov chain Monte Carlo (MCMC) replicates after a burn-in period of 1,000,000. The most likely number of clusters was inferred with the method of *Evanno, Regnaut & Goudet (2005)* implemented in Structure Harvester (*Earl & VonHoldt, 2012*).

Additionally, the microsatellite dataset was analyzed with SAMOVA. We applied the same parameters and criteria that were used in the mtDNA analyses, to keep settings uniform across datasets. Bar plots were created with R to show STRUCTURE and SAMOVA results for each individual. The resulting SAMOVA groupings were used to calculate several population metrics as described in the following. Expected and observed heterozygosity, number of alleles and $F_{ST}$ values between the resulting groups were calculated with Arlequin 3.5. Effective population size was estimated with the coalescent method implemented in NeEstimator v2 (*Do et al., 2014*). Allelic richness and private allelic richness were calculated applying the rarefaction method implemented in ADZE 1.0 (*Szpiech, Jakobsson & Rosenberg, 2008*), with a standardized sample size equal to the smallest sample size across SAMOVA groups, and filtering out loci with more than 50% missing data in any given group. In order to test the effect of missing data on the private alleles and richness calculations, two additional analyses tolerating 25% and 0% missing data were run.

**Table 1 Sources of variation for mtDNA and microsatellite data calculated with SAMOVA under $K = 10$ and $K = 5$, respectively.**

| Marker | Source of variation | d.f. | Sum of squares | Variance components | % variation | Fixation indices |
|---|---|---|---|---|---|---|
| mtDNA | Among groups | 9 | 3,061.309 | 9.779 | 79.28 | $F_{CT} = 0.793$ |
| | Among populations within groups | 43 | 332.944 | 1.002 | 8.12 | $F_{SC} = 0.392$ |
| | Within populations | 291 | 452.013 | 1.553 | 12.59 | $F_{ST} = 0.874$ |
| | Total | 343 | 3,846.266 | 12.334 | | |
| microsatellites | Among groups | 4 | 268.732 | 0.517 | 21.66 | $F_{CT} = 0.217$ |
| | Among populations within groups | 44 | 164.372 | 0.145 | 6.08 | $F_{SC} = 0.078$ |
| | Among individuals within populations | 292 | 513.913 | 0.034 | 1.42 | $F_{IS} = 0.02$ |
| | Within individuals | 341 | 577 | 1.692 | 70.84 | $F_{IT} = 0.292$ |
| | Total | 681 | 1,524.018 | 2.389 | | |

Note:
Bold font indicates statistically significant values ($p < 0.05$); d.f., degrees of freedom

The software NewHybrids (*Anderson & Thompson, 2002*; *Anderson, 2008*) was used to calculate hybrid indices between the SAMOVA defined genotypic clusters. This method employs a Bayesian model in which parental and various classes of hybrids form a mixture from which the sample is drawn. Throughout the manuscript we apply the terminology used by NewHybrids when referring to "hybrid" categories and indices calculated with this software. However, the individuals in this study are admixed individuals but not necessarily inter-specific "hybrids" (i.e., resulting from inter-species mating) as intended in NewHybrids.

We estimated the posterior probability $P(z)$ that each individual in a pair of clusters ($X$ and $Y$) falls into each of six hybrid classes: pure cluster $X$, pure cluster $Y$, $F_1$, $F_2$, cluster $X$ backcross, or cluster $Y$ backcross. Five independent MCMC analyses were run for each pair of neighboring clusters with at least 300,000 iterations after 10,000 burnin sweeps. To evaluate if the MCMC reached convergence, we observed the NewHybrids graphical output and visually assessed whether the complete-data log-likelihood trace increased and stabilized in a parameter space region. $P(z)$ values were averaged among the five independent runs. An individual was considered as belonging to a given class if it is assigned with $P(z) > 0.8$ (*Anderson & Thompson, 2002*).

## RESULTS

### Mitochondrial data

Out of the 368 individuals included in this study, 344 were sequenced for a fragment of the ND4 mtDNA locus. To show the relationships of the two previously unpublished haplotypes KT003209–KT003210, we constructed a haplotype network (Fig. S1). This network constitutes an update from the one produced in 2011 (*Zarza, Reynoso & Emerson, 2011*). The new haplotypes connected to haplotypes in the North A clade, and did not alter the previously observed patterns. In the SAMOVA analyses, a change of less than 1% in FCT was observed at $K = 10$ (Table A and Fig. A in File S2). Under this $K$, 79% of the variation can be explained by variation among groups (Table 1). These groups (mt1–mt10 from now onward, Fig. S2) coincide almost entirely with the haplotype

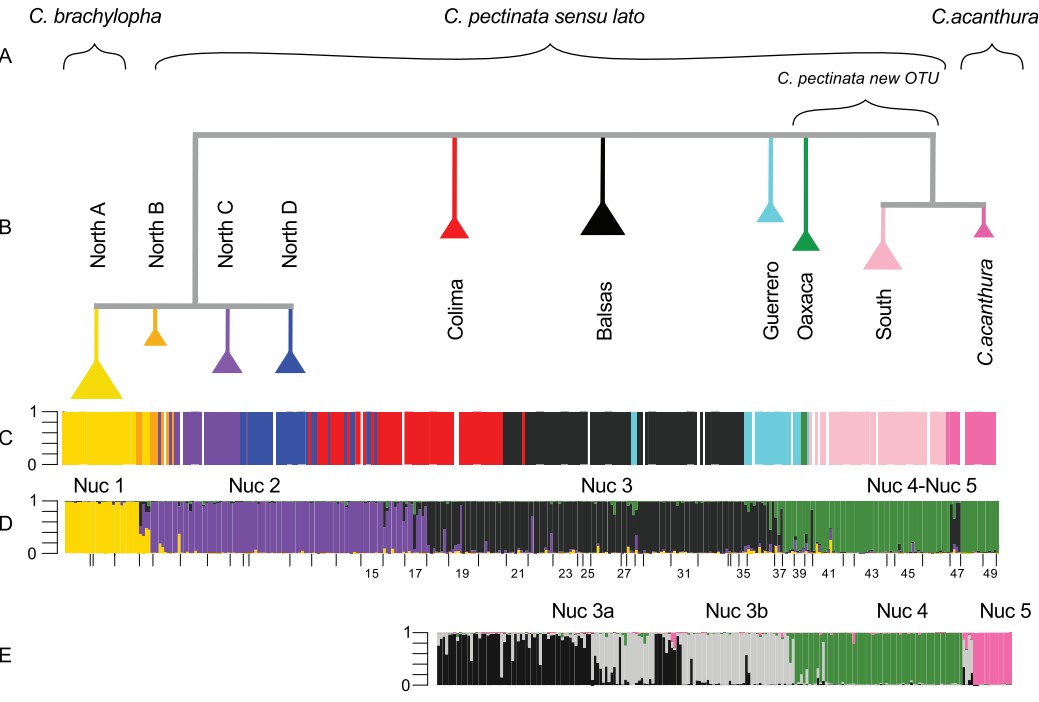

**Figure 2 Phylogenetic tree and population assignment results.** (A) Suggested taxonomic changes in relation to mtDNA and nuclear data analyses; (B) Phylogenetic tree calculated with mtDNA sequences showing the clades proposed by *Zarza, Reynoso & Emerson (2008)*; (C) mtDNA lineage of each individual as defined in haplotype networks calculated by *Zarza, Reynoso & Emerson (2008, 2011)*; (D) microsatellite genotypic cluster defined with STRUCTURE under $K = 4$; (E) substructure estimated with STRUCTURE in a reduced data set (South-SS analyses). In STRUCTURE plots, the *Y*-axis represents proportion of ancestry. Each bar represents an individual. White bars are missing data.

groups previously defined by *Zarza, Reynoso & Emerson (2008)*. In the current study, SAMOVA analyses detected a subdivision (mt4, mt5) within the Colima mtDNA lineage not found in previous studies. Similarly, individuals forming the North B mtDNA lineage (*Zarza, Reynoso & Emerson, 2008*), were here assigned to two different groups (mt1 and mt2). The Oaxaca mtDNA lineage was not recovered in the SAMOVA analyses (Fig. 2).

## Microsatellite data

We obtained genotypes for 341 individuals from 49 localities. Number of samples per locality ranged from one to 15 (File S1). Locus Pec25 suffered from null alleles at a frequency higher than 20% in 12 localities, thus it was not included in further analyses. Other loci are possibly affected by null alleles but in less than 10% of the localities, which may reflect local phenomena leading to homozygous excess (*Chapuis & Estoup, 2007*). The remaining loci exhibited 9–27 alleles among the sampled localities. The null hypothesis of random union of gametes was rejected in 12 localities, but only in one location (La Fortuna, see File S1) was deviation from Hardy–Weinberg equilibrium detected in more than one locus (Pec01, Pec03). After Šidák correction ($p < 0.00007$), the null hypothesis of independence of genotypes at one locus from genotypes at another

Table 2 Summary statistics per locus for genotypic clusters (*Nuc 1–*Nuc 5) defined with SAMOVA.

| | *Nuc 1 | | | | *Nuc 2 | | | | *Nuc 3 | | | | *Nuc 4 | | | | *Nuc 5 | | | |
|---|---|---|---|---|---|---|---|---|---|---|---|---|---|---|---|---|---|---|---|---|
| $L$ | $A$ | $H_O$ | $H_E$ | $F_{IS}$ | $A$ | $H_O$ | $H_E$ | $F_{IS}$ | $A$ | $H_O$ | $H_E$ | $F_{IS}$ | $A$ | $H_O$ | $H_E$ | $F_{IS}$ | $A$ | $H_O$ | $H_E$ | $F_{IS}$ |
| 1 | 4 | 0.48 | 0.49 | 0.01 | 13 | 0.86 | 0.86 | 0.00 | 12 | 0.79 | 0.82 | 0.05 | 14 | 0.68 | 0.84 | 0.19 | 3 | 0.14 | 0.52 | 0.73 |
| 2 | 2 | 0.41 | 0.50 | 0.19 | 6 | 0.33 | 0.35 | 0.05 | 7 | 0.44 | 0.45 | 0.01 | 6 | 0.36 | 0.49 | 0.27 | 4 | 0.14 | 0.51 | 0.72 |
| 3 | 8 | 0.85 | 0.80 | −0.06 | 15 | 0.78 | 0.85 | 0.08 | 25 | 0.83 | 0.91 | 0.09 | 13 | 0.54 | 0.79 | 0.32 | 5 | 0.64 | 0.75 | 0.15 |
| 4 | 2 | 0.52 | 0.50 | −0.03 | 7 | 0.45 | 0.45 | 0.00 | 8 | 0.74 | 0.80 | 0.07 | 5 | 0.14 | 0.16 | 0.14 | 3 | 0.64 | 0.63 | −0.02 |
| 5 | 3 | 0.78 | 0.68 | −0.15 | 7 | 0.51 | 0.55 | 0.06 | 6 | 0.25 | 0.28 | 0.10 | 5 | 0.63 | 0.71 | 0.13 | 2 | 0.07 | 0.07 | 0.00 |
| 6 | 3 | 0.15 | 0.14 | −0.05 | 11 | 0.77 | 0.84 | 0.09 | 10 | 0.50 | 0.66 | 0.25 | 11 | 0.72 | 0.84 | 0.14 | 2 | 0.29 | 0.48 | 0.41 |
| 7 | 6 | 0.59 | 0.62 | N.A. | 10 | 0.63 | 0.74 | N.A. | 14 | 0.55 | 0.74 | N.A. | 4 | 0.07 | 0.58 | 0.10 | m | m | m | N.A. |
| 8 | 8 | 0.67 | 0.82 | 0.19 | 13 | 0.77 | 0.80 | 0.04 | 13 | 0.83 | 0.86 | 0.03 | 8 | 0.67 | 0.74 | | m | m | m | N.A. |
| $M$ | 4.5 | 0.56 | 0.57 | | 10.3 | 0.64 | 0.68 | | 11.9 | 0.62 | 0.69 | | 8.3 | 0.48 | 0.65 | | 3.2 | 0.32 | 0.49 | |
| s.d. | 2.5 | 0.22 | 0.22 | | 3.3 | 0.19 | 0.20 | | 6.0 | 0.21 | 0.22 | | 3.9 | 0.26 | 0.23 | | 1.2 | 0.26 | 0.23 | |
| $n$ | 27 | | | | 105 | | | | 131 | | | | 64 | | | | 14 | | | |
| AR | 3.29 (2.64) | | | | 5.14 (3.98) | | | | 5.54 (4.65) | | | | 4.7 (3.19) | | | | 2.56 (1.38) | | | |
| PA | 0.29 (0.07) | | | | 0.62 (0.18) | | | | 0.96 (0.60) | | | | 0.76 (0.40) | | | | 0.53 (0.29) | | | |
| Ne | 35.1 (0–176) | | | | 8.5 (1.8–20.5) | | | | 15.8 (5.8–30.8) | | | | 22.5 (0–112.8) | | | | 1.9 (1.3–2.7) | | | |

Note:
$L$, Locus; $A$, Allele number; $H_O$, Observed heterozygosity; $H_E$, Expected heterozygosity; $F_{IS}$, inbreeding coefficient; N.A., missing data; m, monomorphic locus; $n$, number of individuals; $M$, Mean; s.d., standard deviation; AR, Allele richness; PA, Private alleles mean (variance); Ne, Effective population size (Jackknife CI).

locus could not be rejected. Pairwise $F_{ST}$ values showed a wide range of genetic differentiation among localities, from non-differentiation ($F_{ST} = 0$) to a high degree of differentiation (maximum significant $F_{ST} = 0.66$; Table S1).

SAMOVA analyses with microsatellite data showed a FCT change <1% under $K = 5$ (from now onward *Nuc 1–*Nuc 5; Table A and Fig. B in File S2). Under this scenario, around 22% of the variation is explained by variation among groups, whereas 71% of the variation was explained by variation within individuals (Table 1). These clusters differ from the mtDNA grouping schemes obtained with SAMOVA, but coincide with the clustering resulting from the STRUCTURE analysis as explained below. Allele number, observed heterozygosity, expected heterozygosity, inbreeding coefficient, and effective population size for *Nuc 1–*Nuc 5 are shown in Table 2. The standardized sample size for the allelic and private allelic richness was 14. Locus Cthe12 was removed from these calculations because it had at least one grouping (i.e., groupings 4 and 5) with more than 50% missing data. Allelic and private richness mean and variance values are shown in Table 2. Genetic differentiation ($F_{ST}$ values) between the SAMOVA groups is shown in Table 3.

STRUCTURE results suggest that the most likely number of genotypic clusters is seven, based on the Delta-$K$ ($\Delta K$) value. However we suspect that $\Delta K$ under $K = 7$ is an artifact resulting from the large variation in likelihood values obtained with the previous $K$, $K = 6$ (SD = 1,231.92; Fig. C in File S2). After removing two runs that seemed to be outliers due to lower likelihood values, the SD under $K = 6$ was greatly reduced (90.45). We then recalculated $\Delta K$. This time $K = 4$ showed the highest $\Delta K$ (Fig. D in File S2). Individuals were consistently assigned among runs. However these results differ from the

**Table 3 Differentiation between SAMOVA clusters ($F_{ST}$ values) estimated with Arlequin 3.5.**

|  | *Nuc 1 | *Nuc 2 | *Nuc 3 | *Nuc 4 |
|---|---|---|---|---|
| *Nuc 1 | 0 |  |  |  |
| *Nuc 2 | 0.18952 | 0 |  |  |
| *Nuc 3 | 0.26536 | 0.14815 | 0 |  |
| *Nuc 4 | 0.28768 | 0.18468 | 0.15797 | 0 |
| *Nuc 5 | 0.44634 | 0.36999 | 0.32849 | 0.34052 |

Note:
All values are statistically significant ($p < 0.05$).

clustering obtained with SAMOVA where more groupings were detected in the southern part of the distribution resulting in $K = 5$ (Fig. S2). Additionally, the SAMOVA analyses detected the separation of *C. acanthura* from *C. pectinata*, whereas STRUCTURE lumped *C. acanthura* with southern populations of *C. pectinata*. Thus to establish the most likely number of $K$ in the southern part of the distribution, and to test for potential equivalents with the SAMOVA analyses and known taxonomy, further analyses were performed on a subset of individuals that included only iguanas collected south of Manzanillo (M in Fig. 1) and along the Gulf of Mexico. We refer to these analyses as South-SS from now onward. Simulations for 10 million generations were run with $K = 2$–$K = 6$, with 10 replicates each. $K = 4$ showed the highest $\Delta K$ with consistent results among runs (Fig. E in File S2). When analyzing the entire dataset, only one cluster was detected between Manzanillo and Las Negras (between M and N in Fig. 1; Nuc 3 in Fig. 2D), whereas two clusters were recognized in the South-SS analyses (Nuc 3a and Nuc 3b in Fig. 2E). However, several individuals of Nuc 3a and 3b showed admixed ancestry, indicating weak geographic structure (Fig. 2E). The division between Nuc 3a and 3b was not detected with SAMOVA (Fig. S2). Two other clusters were identified with the South-SS analyses, one equivalent to *Nuc 4 and the other comprising individuals identified as *C. acanthura* and equivalent to *Nuc 5 (Fig. S2). Individuals forming these two clusters were consistently assigned among runs and in accordance with the assignment observed when analyzing the entire dataset.

Given the weak geographic structure observed between Nuc 3a and Nuc 3b and the lack of support for such subdivision with SAMOVA, we take a conservative approach and consider these as forming only one genotypic cluster (equivalent to *Nuc 3 and Nuc 3). Both SAMOVA and STRUCTURE support the distinction between *Nuc 4 (Nuc 4) and *Nuc 5 (Nuc 5, in the South-SS analyses). Taking into account the results of SAMOVA and STRUCTURE we recognize a total of five microsatellite genotypic clusters within the entire distribution of *C. pectinata* + *C. acanthura* (Figs. 1 and 2).

The microsatellite genotypic clusters detected with STRUCTURE (Nuc 1–Nuc 5) and SAMOVA (*Nuc 1–*Nuc 5) are geographically localized (Fig. 1). The limits of the clusters defined with SAMOVA appear sharp, as this algorithm does not take admixture into account. However, the presence of introgression is supported by the hybrid indices calculated with NewHybrids between SAMOVA genotype clusters (Table 4). Sharp limits

**Table 4 Number of individuals assigned to each hybrid class according to NewHybrids.**

| X,Y | Pure *Nuc X | Pure *Nuc Y | $F_1$ | $F_2$ | *Nuc X Bc. | *Nuc Y Bc. | Un-assigned | n (X + Y) |
|---|---|---|---|---|---|---|---|---|
| *Nuc 1,*Nuc 2 | 26 | 92 | 0 | 1 | 0 | 0 | 13 | 132 |
| *Nuc 2,*Nuc 3 | 83 | 0 | 0 | 37 | 0 | 4 | 112 | 236 |
| *Nuc 3,*Nuc 4 | 110 | 56 | 0 | 0 | 2 | 0 | 27 | 195 |
| *Nuc 3,*Nuc 5 | 125 | 14 | 0 | 2 | 0 | 0 | 4 | 145 |
| *Nuc 4,*Nuc 5 | 14 | 64 | 0 | 0 | 0 | 0 | 0 | 78 |

**Note:**
In all cases, SAMOVA defined clusters were compared.
X,Y = SAMOVA-defined Genotypic cluster compared. As in the main text, tables and figures, the *Nuc prefix denotes SAMOVA defined genotypic cluster.
Bc, backcross.

of clusters are not observed among the genotypic clusters defined with STRUCTURE but admixed individuals and zones of overlap are clearly observed (Figs. 1 and 2).

There are different levels of geographic concordance between the distribution of mtDNA lineages North A, North B, North C, North D, Colima, Balsas, Guerrero, Oaxaca and South as described by Zarza, Reynoso & Emerson (2008, 2011) and genotypic clusters (Figs. 1 and 2). In northern Mexico, the distributions of genotypic cluster Nuc 1 (and *Nuc 1) and the North A mtDNA lineage are almost entirely concordant. Further south, in Central Mexico, Nuc 1 overlaps with Nuc 2. Most of the samples in the SAMOVA-equivalent genotypic clusters (*Nuc 1 and *Nuc 2) were assigned to a "pure" category with NewHybrids (Table 4). Only one F2 was detected and 13 individuals could not be assigned to any category. However four of these individuals had a posterior probability <0.2 of being a "pure" individual. Thus, given the data and the assumptions of the model, those four individuals have a posterior probability >0.8 of being hybrids of some sort. Indeed, STRUCTURE plots show signs of interbreeding in the contact zone (Fig. 2D).

Individuals forming Nuc 2 have mtDNA haplotypes belonging to North A, North B, North C, North D and Colima mtDNA lineages. Genotypic cluster Nuc 2 forms a contact zone with Nuc 3. Individuals in this last cluster carry mtDNA haplotypes of Colima, Balsas and Guerrero lineages. The geographically discordant patterns between mtDNA (North C–D, Colima, Balsas) and microsatellite markers in this area (Nuc 2 and Nuc 3) have been previously detected and described (Zarza, Reynoso & Emerson, 2011).
In the equivalent SAMOVA clusters, 83 individuals were assigned to *Nuc 2 pure class. Pure individuals of *Nuc 3 were not found, however 37 and 4 individuals were assigned to the F2 and *Nuc 3 backcross hybrid classes, respectively (Table 4). Almost 50% of the individuals forming these clusters could not be assigned to any category. Among these, 83 individuals showed a posterior probability <0.2 of belonging to any of the pure classes, thus they might be hybrids of some sort. $F_{ST}$ values between these genotypic clusters are the lowest observed in the pairwise comparisons (Table 3).

Genotypic cluster Nuc 3 overlaps with Nuc 4, which is formed by individuals collected in southeast Mexico with mtDNA haplotypes belonging to the Guerrero, Oaxaca and South mtDNA lineages. Most of the individuals were assigned to one of the pure categories in the SAMOVA equivalents *Nuc 3 and *Nuc 4 (Table 4). Only two *Nuc 3 back-crosses

were found and 27 were not assigned to any category. None of them had posterior probability <0.2 of belonging to any pure class.

Nuc 4 and Nuc 5 do not overlap. All individuals in the SAMOVA equivalents *Nuc 4 and *Nuc 5 were assigned to a pure category with a posterior probability >0.8. Nuc 5 includes individuals described as *C. acanthura*, collected in eastern Mexico. It is geographically concordant with the distribution of a mtDNA lineage closely related to the Southern mtDNA lineage (*Zarza, Reynoso & Emerson, 2008*). Admixture between *C. acanthura* and *C. pectinata* is only evident in Zapotitlán de las Salinas (denoted as "Z" in Fig. 1), with individuals carrying *C. acanthura* mtDNA haplotypes but with nuclear ancestry of Nuc 3 and Nuc 5. The NewHybrids analysis between *Nuc 3 and *Nuc 5 detected two F2 individuals. One was collected in Zapotitlán de las Salinas, and the other in Apatzingán (denoted as "Z" and "A," respectively in Fig. 1). The latter locality is not geographically close to the distribution limits of Nuc 5 (or *Nuc 5). Thus the potential of long distance dispersal, perhaps human mediated, should be investigated. The remaining of the individuals was assigned to one of the pure categories and only four were not assigned to any hybrid or pure category.

# DISCUSSION

## Introgression and geographic discordance between mtDNA and nuclear markers

Different degrees of discordance are observed in the geographic distribution of mtDNA lineages and microsatellite genotypic clusters across the range of *C. pectinata*. At one end of the spectrum, mtDNA North A lineage is almost entirely concordant with Nuc 1 cluster. At the opposite end of the spectrum, mtDNA lineages distributed along the central western coast of Mexico exhibit a striking discordant pattern where up to six geographically distinct mtDNA lineages (North A, North B, North C, North D, Colima, Balsas) co-occur with only two nuclear clusters (Nuc 2 and Nuc 3). This discordance between maternally and biparentally inherited markers in *C. pectinata* might be the result of several processes acting alone, in concert or at different points in time. For example, a suitable explanation might be a scenario of short term refugia where populations decline throughout the range, resulting in isolation, followed by recent range expansion and male biased dispersal (*Dubey et al., 2008*; *Johansson, Surget-Groba & Thorpe, 2008*; *Ujvari, Dowton & Madsen, 2008*; *Zarza, Reynoso & Emerson, 2011*; *Toews & Brelsford, 2012*). The discordant pattern can also be the result of coalescence stochasticity (*Irwin, 2002*; *Hickerson et al., 2010*), selection of mtDNA (*Dowling, Friberg & Lindell, 2008*), or differences in effective population size between mtDNA and nuclear markers.

Introgression, along current and past contact zones, may have also contributed to the patterns of geographic discordance in conjunction with other demographic phenomena. For example, it has been suggested that, in contact zones, selection and genetic drift can lead to mtDNA introgressing further and faster than nDNA. This is because mitochondrial genomes are less likely to hitchhike with a region under selection that prevents introgression (*Ballard & Whitlock, 2004*; *Petit & Excoffier, 2009*; *Milá et al., 2013*). Additionally, in small populations, genetic drift can allow the fixation of

slightly deleterious alleles in the mtDNA of one population resulting in lower fitness than a related species in the same area. Selection could then drive introgression of mtDNA from the more fit population into the less fit population (*Ballard & Whitlock, 2004*). Furthermore, it is possible that some contact zones have changed location (*Barton & Hewitt, 1985*; *Buggs, 2007*), or that others have disappeared entirely as a result of complex climate mediated cycles of range expansion and contraction, or due to other phenomena. It is difficult to disentangle the effect of these processes with the currently available data. Sampling more finely along contact zones, and sequencing additional nuclear markers may permit coalescence analyses (*Singhal & Moritz, 2012*). Behavioral studies may also be informative to evaluate the effects of ecological, demographic, historical and stochastic factors shaping the discordant patterns.

Interestingly, pairs of inter-breeding nuclear clusters with different levels of divergence occur throughout the distribution of *C. pectinata*. For example, allele frequency divergence between *Nuc 1 and *Nuc 2 is 0.18952, whereas it is 0.14815 between *Nuc 2 and *Nuc 3 (Table 3). Assignment of individuals to pure and hybrid classes also shows that contact zones have different hybrid compositions. A higher proportion of individuals were assigned to a pure class when analyzing *Nuc 1 and *Nuc 2 (89%) than when analyzing *Nuc 2 and *Nuc 3 (36%). This is also observed in the STRUCTURE plots which reveal Nuc 2 and Nuc 3 admixed individuals more frequently than admixed Nuc 1 and Nuc 2.

Thus *C. pectinata* constitutes an excellent system to better understand the process of speciation by studying the effects of introgression between genotypic clusters at different stages of divergence. Furthermore, this system potentially enables the comparison of evolutionary patterns and processes with contact zones in temperate and other tropical regions of the world (*Leaché & McGuire, 2006*; *McGuire et al., 2007*; *Singhal & Moritz, 2012*; *Miraldo et al., 2013*; *Milá et al., 2013*).

## Implications for *Ctenosaura pectinata* taxonomy and conservation

Our results suggest that there are five nuclear genotypic clusters forming what is currently considered *C. pectinata*. Individuals forming the Nuc 1 cluster belong to the North A mtDNA lineage. Thus Nuc 1 and North A mtDNA lineages are geographically concordant. The distribution of this genotypic cluster coincides with the distribution of *C. brachylopha* as revised by *Bailey (1928)* using morphological data (i.e., states of Sinaloa, Nayarit, North of Jalisco and Isla Isabel; Figs. 1 and 2).

The observed concordance in the geographic distribution of nuclear and mtDNA might be the result of stochastic coalescent processes, which is particularly true in taxa with low dispersal rates, as is the case for iguanas (*Irwin, 2002*). Other phenomena such as natural selection could be shaping the observed pattern, however this cannot be evaluated with the currently available data. Another possibility is that the formation of a biogeographic barrier affected the distribution of Nuc 1 and North A. Their southern distribution limit coincides approximately with the Trans-Mexican Volcanic Belt (TMVB). This geographic feature has been proposed as a geographic barrier for several lowland taxa

(*Devitt, 2006*; *Mulcahy, Morrill & Mendelson, 2006*; *De-Nova et al., 2012*; *Arbeláez-Cortés, Milá & Navarro-Sigüenza, 2014*; *Arbeláez-Cortés, Roldán-Piña & Navarro-Sigüenza, 2014*; *Suárez-Atilano, Burbrink & Vázquez-Domínguez, 2014*; *Blair et al., 2015*). However, given the complex geological history of the area, the TMVB barrier might not have affected all taxa equally (*Mastretta-Yanes et al., 2015*). Indeed, despite this barrier, gene flow has occurred between Nuc 1 and the neighboring Nuc 2 at the limits of their distribution in the vicinity of the TMVB.

Gene flow has also been observed in a contact zone between Nuc 1 and *C. hemilopha* in the northern edge of Nuc 1 distribution (*Zarza, Reynoso & Emerson, 2008*). Along both, northern and southern edges, gene flow seems to be limited to a narrow area. According to hybrid zone theory, several factors affect the extent, maintenance and shifting of hybrid zones: dispersal, selection, recombination rates and time since secondary contact (*Barton & Hewitt, 1985*). The effect of these processes needs to be further investigated, ideally at the genomic level.

The paradigm that lack of gene flow is a prerequisite to maintain species integrity is shifting (*Abbott et al., 2013*). In recent years evidence has accumulated suggesting that gene flow is an integral part of the process of speciation and that divergence can occur in the presence of gene flow (*Mallet, 1995*; *Pinho & Hey, 2010*; *Leaché et al., 2014*; *Zarza et al., 2016*; *Leavitt et al., 2017*). Indeed, if reproductive barriers have emerged in secondary contact zones, it is uncertain whether barriers to gene flow will be strengthened or broken down due to recombination and admixture (*Barton & Hewitt, 1985*; *Abbott et al., 2013*).

Despite the levels of gene flow detected and given the geographic concordance in the distribution of mtDNA and nuclear markers, the geographic limits that coincide with the geographic limits of other species, and the morphological signal detected by *Bailey (1928)*, we suggest the resurrection of the name *C. brachylopha* for populations inhabiting northwestern Mexico.

The distribution of Nuc 2 and Nuc 3 genetic clusters are geographically discordant with the distribution of mtDNA lineages in central Mexico (North A–D, Colima, Balsas). Maternal lineages are more deeply structured than the genotypic clusters. The distribution of the maternally and paternally inherited markers and the high number of sampled admixed individuals suggest that, although there is some substructure in the area, gene flow among populations has been ongoing. Given that the holotype locality is labeled as "Colima" (*Wiegmann, 1834*) we suggest that these genotypic clusters keep the historical name *C. pectinata* (Fig. 2).

Iguanas described as *C. acanthura* also form a coherent nuclear cluster (Nuc 5) that is concordant with a mtDNA lineage closely related to the South clade (*Zarza, Reynoso & Emerson, 2008*). Thus the name *C. acanthura* should continue to be applied to populations of spiny-tailed iguanas in the coast of the Gulf of Mexico. Introgression is apparent in the area of Zapotitlán de las Salinas (Fig. 1), where individuals carry mtDNA haplotypes typical of *C. acanthura* and some alleles of Nuc 3 and Nuc 5.

Nuc 4 is almost entirely geographically concordant with the South mtDNA lineage, but also overlaps with the Oaxaca and Guerrero lineages. Thus Nuc 4 deserves taxonomic recognition at the species level, and awaits full description until

morphological data is gathered and analyzed. In the meantime, we propose that these populations are recognized as an independent Evolutionary Significant Unit (*Moritz, 1994*) within *C. pectinata*.

We are aware that the modifications in taxonomy proposed in this paper are based mostly on molecular and geographic evidence. Morphological data have not revealed the existence of divisions within *C. pectinata* (*Köhler, Schroth & Streit, 2000*), except for the work of Bailey (*Bailey, 1928*). He realized that *C. brachylopha* resembles *C. pectinata* but may be distinguished from it by having a median dorsal crest that does not extend over the sacral region and that it is formed by 65–75 scales. He also noticed that the first seven whorls of spinous caudal scales are separated from each other by three rows of small flat scales. In *C. pectinata* the first five whorls of spinous scales are separated from each other by three rows of small flat scales, but subsequent whorls of spinous scales are separated by two rows of flat scales up the middle of the length of the tail (*Bailey, 1928*). These and other morphological characters need to be studied in depth, with a large sample and with more modern statistical methods to validate their utility to distinguish between groups within *C. pectinata*. Color may be an important character too. Individuals from northern Mexico exhibit a yellow coloration (Fig. S3), those in central Mexico show blue and orange patterns, and individuals from the south are black and white. Bailey studied stuffed or alcohol-preserved specimens that most likely lost their original color, so he did not address this character.

Our molecular approach has uncovered several genotypic clusters. However this may present challenges for the field biologist working in areas with high levels of admixture (i.e., central western Mexico) and with only morphological data at hand. Further research is needed to determine if coloration patterns or morphological characters of individuals outside the contact zones provide information for their assignment to a specific genotypic cluster.

This work provides important knowledge with profound implications in conservation, wildlife management and forensics. *Ctenosaura pectinata sensu lato* faces illegal hunting, poaching and habitat loss (*Reynoso, 2000*; *Aguirre-Hidalgo, 2007*; *Faria et al., 2010*). It is considered as a threatened species under the Mexican law (*SEMARNAT, 2002*), awaiting IUCN evaluation, and may not receive proper protection if its genetic composition and distribution is not taken into account (*Frankham, 2006*). Measures have been taken to protect its populations, however there are no genetics-based strategies to re-introduce confiscated individuals and/or their offspring. Ideally, the genetic origin of iguanas should be recognized before re-introduction to avoid admixture in populations that may lead to loss of diversity through hybridization, reduced viability or fertility in the case of genetic incompatibilities, reduced population fitness due to selective disadvantage of intermediate genotypes or loss of advantageous parental traits (*Lynch, 1991*; *Burke & Arnold, 2001*). Furthermore, our results suggest that *C. pectinata*, a species already recognized as threatened, is actually composed of multiple genotypic clusters that might be at a higher risk than previously thought, given their reduced geographical distributions and effective population sizes (*Bickford et al., 2007*).

## ACKNOWLEDGEMENTS

EZ and VHR would like to dedicate this work to the memory of Wendoli Medina Mantecón, a close collaborator and friend who will be remembered, among other things, for her efforts and dedication to the conservation of iguanas and other endangered species.

### Funding

The Mexican Council for Science and Technology (CONACYT) funded this research through a PhD grant to Eugenia Zarza at the University of East Anglia, where the data was generated. The funders had no role in study design, data collection and analysis, decision to publish, or preparation of the manuscript.

### Grant Disclosure

The following grant information was disclosed by the authors:
The Mexican Council for Science and Technology (CONACYT).

### Competing Interests

The authors declare that they have no competing interests.

### Author Contributions

- Eugenia Zarza conceived and designed the experiments, performed the experiments, analyzed the data, contributed reagents/materials/analysis tools, prepared figures and/or tables, authored or reviewed drafts of the paper, approved the final draft.
- Victor H. Reynoso conceived and designed the experiments, contributed reagents/materials/analysis tools, prepared figures and/or tables, authored or reviewed drafts of the paper, approved the final draft.
- Christiana M. A. Faria performed the experiments, analyzed the data, authored or reviewed drafts of the paper, approved the final draft.
- Brent C. Emerson conceived and designed the experiments, contributed reagents/materials/analysis tools, authored or reviewed drafts of the paper, approved the final draft.

### Animal Ethics

The following information was supplied relating to ethical approvals (i.e., approving body and any reference numbers):

The School of Biological Sciences Ethical Review Committee at the University of East Anglia approved this study.

### Field Study Permissions

The following information was supplied relating to field study approvals (i.e., approving body and any reference numbers):

Blood samples were obtained under the permits SEMARNAT SGPA/DGVS/08239, SGPA/DGVS/02934/06, 03563/06 approved by the Minister of environment and natural resources (Secretaría del Medio Ambiente y Recursos Naturales) to VHR.

## DNA Deposition

The following information was supplied regarding the deposition of DNA sequences:

Sequence data can be found at GenBank using accession numbers KT003209–KT003210.

## Data Availability

The raw data is available in File S1.

## Supplemental Information

Supplemental information for this article can be found online at http://dx.doi.org/10.7717/peerj.6744#supplemental-information.

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
