# Peer review of "Introgressive hybridization in a Spiny-Tailed Iguana, Ctenosaura pectinata, and its implications for taxonomy and conservation"

_PeerJ, doi:10.7717/peerj.6744_

## Round 0.1 · original submission · Major Revisions

Dear Authors,

Thank you for submitting a fine MS to PeerJ. Both referees suggested minor revision, however, based on the extent of the revisions suggested by the first reviewer and my reading of your MS, I recommend major revision.

In addition to the comments of the reviewers, and in support of them, you need to:

1) be clear on how lineages were delimited
2) on how you used the microsat data, and what kind of information these data contain (you can infer the existence of biological groups with one or more biological groups comprising an evolutionary lineage)
3) the proposed taxonomy should be clearly seen on the phylogenetic trees/hypotheses
4) the results and interpretation of results of admixture/introgression should be more explicit and visualized better, i.e. more easily accessible for the readers
5) edit the MS for grammar and structure

Otherwise I think you did a nice job, and I look forward to seeing a revised version.

Sincerely,

Tomas Hrbek

Reviewer 1 ·

Basic reporting

This manuscript describes genetic clustering patterns in iguanas in Mexico, and the results have taxonomic implications for the species. The study provides both mitochondrial DNA sequences and microsatellites.
The conceptual basis of the study is to compare clustering patterns between nuclear and mtDNA data, and to quantify introgression. This is accomplished by 1) defining genotypic clusters, 2) examining concordance between markers, 3) examining introgression levels, and 4) recommendation of taxonomic changes.
The study meets these goals, but the results are not always clear, which leaves the conclusions open to interpretation.

Experimental design

1) I got lost trying to figure out how lineages are diagnosed when no lineage-based approaches are used in this study. Genotype clusters are not lineages. I recommend adding a phylogenetic analysis to this study to provide a direct and transparent avenue for discussing putative lineages. Unfortunately, microsatellites are not well-suited for estimating phylogenetic trees, but the mtDNA data are available. A single locus analysis leaves a lot to be desired, but it is better to have one a one-locus gene tree versus no phylogeny at all

2) Provide an ethics statement and cite formal documents approving animal care and use, and scientific collecting permissions from government agencies.

3) The haplotype network is not a phylogenetic estimate. Replace and/or augment the supplemental figure with a phylogeny for the same data. I realize that the same analysis is likely published in one of the previous studies of the same data, but it is easier to include the phylogeny here than to require readers to find the information in a previous paper.

4) The analysis of the microsatellite data is rather confusing. The description of the Structure analyses is unclear, and it is doubtful that the study could be replicated. Some people use “hierarchical” analyses to dissect Structure results, and it seemed like that might be what these authors are doing, but it is not clear. I was left with the impression that the microsatellite data might support more populations but that the analyses were insufficient to make a firm determination, therefore the authors picked their favorite value, and then performed additional analyses for some groups and not others. The haphazard nature of these analyses should be replaced with a clear and easy-to-follow description of the methodology.

5) The accounting of samples is confusing. It seems like this study only adds two new samples to the existing mtDNA dataset. Is this accurate? If so, there needs to be a justification for why these two new samples merit additional reanalysis. Are they from a geographic area of particular interest? Are they from individuals that appear to be hybrids? What motivates the publication of a study that increases sampling from 342 to 344 specimens, which is a negligible increase?
Finally, the influence of the two new samples on the phylogeny/ clustering/introgression analysis is not stated clearly in the results. Did these two new samples expose something new or interesting in this study that is worth reporting?

6) The proposition of taxonomic changes without illustrating the resultant changes on a phylogenetic tree is highly problematic. I recommend annotating a phylogenetic tree with current and proposed taxonomic changes. These changes need to be transparent and identifiable on a phylogeny. The discussion needs to address taxonomic changes in the context of a phylogeny and not genotypic clusters, which are not equivalent to evolutionary lineages.

7) Is it necessary to include copyright information for GoogleEarth/GoogleMaps images, which the authors seem to have masked in the figures? PeerJ staff should have an answer for this.

8) I found it difficult to identify introgressed/admixed/hybrid/backcrossed individuals on the figures, because this information is never compared directly in a single figure. The maps convey important information, but it is difficult to tell which samples are the same in the different maps, because they have different sampling. Figure three provides a side-by-side comparison, but the this figure does not include a detailed geographic component (only group names). It would be useful to see a mtDNA gene tree with structure/microsat. results plotted on the tips of the tree. Alternatively, it would be useful to plot the microsat structure groups on a map, and then superimpose the mtDNA groups/clades. This would make it much easier to identify areas of concordance and/or discordance.

Validity of the findings

Minor issues:
- Supplant the definition of introgression in the introduction with one found in the population genetics literature. The use of a definition from a recent and somewhat random empirical study seems awkward.
- A recent study of Xenosaurus lizards by Nieto Montes de Oca in Molecular Phylogenetics and Evolution that used SNP data should be cited in the introduction.
- Scientific names need to be in italic font throughout, check the reference section for errors.

Additional comments

see above.

·

Basic reporting

There are a number of areas where the clarity of the English could be improved. The use of appropriate punctuation (specifically an en dash versus a hyphen) needs to be corrected throughout the manuscript. Also, proper abbreviation of generic names needs to follow appropriate conventions throughout, such that the first occurrence of a generic name in a paragraph is spelled out and any occurrences thereafter are abbreviated. Lastly, there are numerous cases of incomplete or improperly formatted citations.

Generally, the intro and background are well-crafted, though the taxon-specific information needs to be reorganized for clarity.

The figures have a number of problems, specifically Figures 1 and 2. The inset on figure 1 is (1) too small an doesn't make full use of the "dead space" below and to its left. The inset should not cover up what it is depicting. Also, there are numerous places in the manuscript where specific named localities are referenced, yet none of them are depicted on either map. This is very confusing for readers that aren't familiar with the geography of Mexico. As point of convenience and added clarity, I also recommend better depicting the specific sampling localities for which admixture was detected and shown in Figure 3. Indicating on the map/s which sampling localities contained the admixed individuals would be quite helpful.

Experimental design

This study represents original research that is within the scope of the journal. The research question is well-defined and relevant, and fills an identified knowledge gap.

Unfortunately, some of the authors' approaches require further justification. The clustering analyses applied are plagued by subjective thresholds that lack any language of justification. Without justification, one cannot know that the authors fully explored the patterns present in their data.

The methods are described in sufficient detail, and are replicable.

Validity of the findings

The impact and novelty is somewhat assessed in the discussion, specifically with how the results impact conservation of the study organism.

The authors support further studies that address the questions and hypotheses raised in their own manuscript.

The data appear to be robust, statistically sound, and controlled

Additional comments

Generally, this is a well-written manuscript and a very interesting study.

I have commented thoroughly throughout the manuscript where necessary. The manuscript requires a thorough read to address a number of inconsistencies and formatting issues.

The figures are generally well done, but their clarity could definitely be improved.

My only real concern with your analyses is that you seem to have arbitrarily chosen values of K to test, rather than exploring a wider range of values and fully exploring the variation in the data. While this may not prove fruitful in practice, it is vital to ensure that analyses and inferences aren't biased.

---

## Round 0.2 · Minor Revisions

Dear Authors,

Thank you for resubmitting your MS to PeerJ. One of the original reviewers re-reviewed your MS and did an excellent job pointing out a number of issues that still need to be addressed. I looked your MS over as well, but at this point have nothing further to add in addition to Luke Welton’s comments and concerns.

I recommend implementing the suggested changes. One this is done, I believe your MS will be ready for publication.

Sincerely,

Tomas Hrbek

·

Basic reporting

This manuscript continues to be plagued by improper presentation of generic names, which I have tried to note throughout.

Experimental design

no comment

Validity of the findings

no comment

Additional comments

no comments

---

## Round 0.3 · accepted · Accept

Dear Authors,

Thank you for resubmitting your MS to PeerJ. After reading your revisions, I am happy to accept the MS for publication.

In the final version please also correct “… there are not genetics-based strategies to re-introduce confiscated ...” to “… there are no genotype-informed strategies for reintroduction of confiscated ...”. This sentence is on line 499.

Congratulations on a job well done.
Sincerely,

Tomas Hrbek

#